# A Nature-Inspired Nrf2 Activator Protects Retinal Explants from Oxidative Stress and Neurodegeneration

**DOI:** 10.3390/antiox10081296

**Published:** 2021-08-16

**Authors:** Maria Grazia Rossino, Rosario Amato, Marialaura Amadio, Michela Rosini, Filippo Basagni, Maurizio Cammalleri, Massimo Dal Monte, Giovanni Casini

**Affiliations:** 1Department of Biology, University of Pisa, 56126 Pisa, Italy; mariagrazia.rossino@phd.unipi.it (M.G.R.); rosario.amato@biologia.unipi.it (R.A.); maurizio.cammalleri@unipi.it (M.C.); 2Section of Pharmacology, Department of Drug Sciences, University of Pavia, 27100 Pavia, Italy; marialaura.amadio@unipv.it; 3Department of Pharmacy and Biotechnology, University of Bologna, 40126 Bologna, Italy; michela.rosini@unibo.it (M.R.); filippo.basagni2@unibo.it (F.B.); 4Interdepartmental Research Center Nutrafood “Nutraceuticals and Food for Health”, University of Pisa, 56124 Pisa, Italy

**Keywords:** retinal disease, antioxidant enzymes, neuronal death, gliosis

## Abstract

Oxidative stress (OS) plays a key role in retinal dysfunctions and acts as a major trigger of inflammatory and neurodegenerative processes in several retinal diseases. To prevent OS-induced retinal damage, approaches based on the use of natural compounds are actively investigated. Recently, structural features from curcumin and diallyl sulfide have been combined in a nature-inspired hybrid (NIH1), which has been described to activate transcription nuclear factor erythroid-2-related factor-2 (Nrf2), the master regulator of the antioxidant response, in different cell lines. We tested the antioxidant properties of NIH1 in mouse retinal explants. NIH1 increased Nrf2 nuclear translocation, Nrf2 expression, and both antioxidant enzyme expression and protein levels after 24 h or six days of incubation. Possible toxic effects of NIH1 were excluded since it did not alter the expression of apoptotic or gliotic markers. In OS-treated retinal explants, NIH1 strengthened the antioxidant response inducing a massive and persistent expression of antioxidant enzymes up to six days of incubation. These effects resulted in prevention of the accumulation of reactive oxygen species, of apoptotic cell death, and of gliotic reactivity. Together, these data indicate that a strategy based on NIH1 to counteract OS could be effective for the treatment of retinal diseases.

## 1. Introduction

The retina is highly susceptible to increases in reactive oxygen species (ROS) due to its constant exposure to light, high oxygen demand, and high levels of fatty acid oxidation [1]. In physiological conditions, ROS support the normal cellular metabolism, but their uncontrolled increase may generate deleterious effects impairing retinal cell integrity. Therefore, ROS levels are constantly regulated by endogenous defense systems, in which a primary role is played by nuclear factor erythroid 2-related factor 2 (Nrf2) [2,3]. Nrf2 acts as a redox-sensible element whose transcriptional activity regulates multiple genes encoding antioxidant proteins and phase II detoxifying enzymes, including heme oxygenase-1 (HO-1) and NADPH dehydrogenase quinone oxido-reductase 1 (NQO1). The activation of Nrf2 occurs when ROS interact with the cytosolic Nrf2 repressor Kelch-like ECH associated protein 1 (Keap1). In normal conditions, Keap1 causes constant Nrf2 degradation through the ubiquitin-proteasome pathway, but, when ROS accumulate, Keap1 is inhibited and Nrf2 may translocate into the nucleus [4,5].

The oxidative unbalance due to ROS accumulation and/or antioxidant defense depletion induces oxidative stress (OS), which acts as a major trigger of retinal diseases such as age-related macular degeneration (AMD), diabetic retinopathy (DR), and glaucoma [6,7]. Indeed, OS promotes mitochondrial dysfunction and inflammation resulting in retinal neuronal morphological and functional deficits [8,9,10]. Considering that OS is the main cause of neurodegeneration, promising pharmacological strategies aimed at favoring both antioxidant and neuroprotective activities have been tested in the last few years. In this respect, nutraceuticals as polyphenols, carotenoids, saponins, and other natural compounds have increasingly been considered for their ability to counteract OS through ROS scavenging and/or induction of antioxidant enzymes [11,12]. Beyond their antioxidant features, nutraceuticals are easy to administer, affordable, and devoid of side effects if administered in appropriate doses [13]. In particular, curcumin, a yellowish polyphenolic compound extracted from *Curcuma longa*, displays prominent antioxidant effects through direct free radicals scavenging, inhibition of ROS-generating enzymes, or modulation of Nrf2 activity [14,15,16,17,18,19]. The curcumin antioxidant effect has been demonstrated to result in significant neuroprotection in AMD, DR, and glaucoma [20,21,22,23,24]. Nevertheless, the curcumin short half-life, low solubility, and rapid metabolism negatively affect its bioavailability and limit its therapeutic applications [25]. To face these issues, different approaches have been followed, including new formulations, different ways of administration and alternative drug delivery systems [26]. In parallel, hybridization strategies have been performed to boost curcumin’s antioxidant properties. In particular, a new nature-inspired hybrid (NIH), dubbed NIH1 [*S*-allyl (*E*)-3-(3,4-dihydroxyphenyl)prop-2-enethioate], has been synthesized by combining the hydroxycinnamoyl function of curcumin with the mercaptan moiety of garlic-derived diallyl sulfide [27,28], a volatile organosulfur compound that is known to induce the expression of antioxidant enzymes via Nrf2 activation [29]. Therefore, NIH1 may induce Nrf2 activation thanks to interactions promoted by both its moieties. Notably, NIH1 emerged for its ability to activate Nrf2 signaling pathway to a higher extent than curcumin in neuroblastoma cells and in THP1 cells [30,31]. In addition, NIH1 was shown to trigger Nrf2 activation and to also reduce H_2_O_2_-induced ROS in ARPE 19 cells [32].

To test whether NIH1 might be useful to treat retinal diseases, it is essential to gather information about its efficacy in models in which the retinal complexity and the relationships between the retinal cellular elements are maintained. For this reason, herein we used organotypic retinal explants to test the effect of NIH1 on the activation of antioxidant responses and on neuroprotection in response to OS. Interestingly, OS is strictly linked to inflammation [33,34], which results in macroglial activation. In this response, both astrocytes and Müller cells increase the expression of glial fibrillary acidic protein (GFAP), which; therefore, can be used as a sensitive marker for reactive gliosis associated to retinal inflammation and neurodegeneration [35]. A GFAP upregulation both in astrocytes and in Müller cells has been reported in most retinal diseases, including AMD [36], DR [37,38], and glaucoma [39,40]. We highlighted that NIH1 triggers an antioxidant pathway that effectively prevents retinal cell death and glial activation, thus demonstrating that NIH1 may be considered for treatments aimed at preventing and/or treating retinal diseases like DR, AMD, and glaucoma.

## 2. Materials and Methods

### 2.1. Reagents

All chemicals were purchased from Merck/Sigma Aldrich (Darmstadt, Germany) unless otherwise stated.

### 2.2. Ex-Vivo Model

Ex-vivo studies were performed using retinas from 3- to 5-week-old C57BL/6J mice following previously established protocols [41]. The procedures were approved by the Commission for Animal Wellbeing of the University of Pisa (permission number: 0034612/2017) and were in compliance with the ARVO Statement for the Use of Animals in Ophthalmic and Vision Research, the Italian guidelines for animal care (DL 26/14), and the EU Directive (2010/63/EU). The mice were kept in a regulated environment (23 ± 1 °C, 50 ± 5% humidity) with a 12 h light/dark cycle (lights on at 8:00 a.m.). The retinas were dissected in Modified Eagle Medium (MEM), then each retina was cut into 4 fragments, which were placed onto Millicell-CM culture inserts (Merck Millipore, Darmstadt, Germany) with ganglion cells facing up. The inserts were transferred to 6-well tissue culture plates with 1 mL of culture medium (50% MEM, 25% Hank’s buffer salt solution, 25% Dulbecco’s Phosphate Buffered Saline, 25 U/mL penicillin, 25 mg/mL streptomycin, 1 µg/mL amphotericin B, and 200 µM L-glutamine). The explants were incubated at 37 °C with 5% CO_2_. The culture medium was changed every day. According to our previous findings, these culture parameters ensure the maintenance of the retinal microarchitecture and neurochemical characteristics [37,41,42,43].

### 2.3. NIH1 Synthesis and Characterization

NIH1 was synthesized and prepared according to previously published procedures [27,28]. Briefly, tert-butyldimethylsilyl protection of caffeic acid followed by coupling with 2-propene-1-thiol in the presence of N,N’-dicyclohexylcarbodiimide and 4-(N,N-dimethylamino)pyridine and a subsequent treatment with tetrabutylammonium fluoride gave compound NIH1 in an overall 18.7% yield. The detailed characterization of NIH1 by nuclear magnetic resonance (NMR) spectroscopy and electrospray ionization mass spectrometry (ESI-MS) has been reported previously [27]. NMR spectra were recorded at 400 MHz for ^1^H and 100 MHz for ^13^C with a Varian MR 400 spectrometer (Varian Inc, Palo Alto, CA, USA). Chemical shifts were reported in parts per millions (ppm) relative to tetramethylsilane, and spin multiplicities were given as s (singlet), d (doublet) or m (multiplet). 

^1^H NMR (400 MHz, CDCl3) δ 7.51 (d, *J* = 16 Hz, 1H), 7.09 (s, 1H), 7.03 (d, *J* = 8 Hz, 1H), 6.88 (d, *J* = 8 Hz, 1H), 6.55 (d, *J* = 16 Hz, 1H), 5.81–5.91 (m, 1H), 5.26–5.31 (m, 1H), 5.13 (d, *J* = 8 Hz, 1H), 3.66 (d, *J* = 8 Hz, 2H).

^13^C NMR (100 MHz, CDCl3) δ 190.90, 147.02, 144.08, 141.50, 132.96, 127.12, 123.14, 122.52, 118.27, 115.79, 114.88, 31.99.

Direct infusion ESI-MS mass spectrum was recorded with a Waters ZQ 4000 apparatus (Waters S.p.A., Sesto San Giovanni, Italy). MS (ESI-): *m*/*z* 235 [M-H]-.

NIH1 was determined >98% pure by HPLC analysis, which was carried out through HPLC reversed-phase conditions on a Phenomenex Jupiter C18 (150 × 4.6 mm I.D.) column (Phenomenex, Castel Maggiore, Italy), UV detection at λ = 302 nm, a flow rate of 1 mL/min with mobile phase ACN/H_2_O 40:60. Analysis was performed on a liquid chromatograph model PU 2089 PLUS equipped with a 20 μL loop valve and linked to MD 2010 Plus UV detector (Jasco Europe, Lecco, Italy).

### 2.4. Treatments

NIH1 was administered to retinal explants at 5, 15, or 50 µM for 24 h in dose-response experiments, while 50 µM NIH1 was administered for six days to test its long-term effects (Figure 1A). The OS treatment was induced pre-incubating the explants for 24 h with basal culture medium, and then adding 100 µM H_2_O_2_ for the subsequent one day or five days (OS/2d and OS/6d, respectively; Figure 1B). Previous studies established that incubation with 100 µM H_2_O_2_ for five days constitutes a reliable model of OS in mouse retinal explants [41]. To test the antioxidant action of NIH1, the explants were pre-treated with 50 µM NIH1 for 24 h and then incubated with 50 µM NIH1 + 100 µM H_2_O_2_ for the subsequent one day or five days (NIH1 + OS/2d and NIH1 + OS/6d, respectively; Figure 1C). Three independent samples, each constituted by eight retinal explants, were used for each experimental group.

### 2.5. Quantitative Real-Time PCR (qPCR)

At the end of the experimental period, the retinal explants were collected and stored at −80 °C. Total RNA extraction was performed using TRIZOL (Thermo Fisher Scientific, Waltham, MA, USA), then the RNA was resuspended in RNase-free water and quantified by spectrophotometric analysis (BioSpectrometer, Eppendorf, Hamburg, Germany). First-strand cDNA was generated from 1 µg of total RNA using the QuantiTect Reverse Transcription Kit (Qiagen, Hilden, Germany). The qPCR analysis was performed to quantify the expression of Nrf2, HO-1, and NQO1 mRNAs using SYBR green Master Mix on a CFX Connect Real-Time PCR System and CFX manager software (Bio-Rad Laboratories, Hercules, CA, USA). The primers were designed to hybridize to unique regions of the analyzed genes. Ribosomal Protein L13a (RPL13A) was used as a reference gene for mRNA level normalization using the ΔΔCt method. Primer sequences are reported in Table 1.

### 2.6. Western Blotting

Total protein extraction was performed using RIPA Lysis buffer supplemented with protease and phosphatase inhibitor cocktails, while nuclear-cytoplasmatic extraction was accomplished using the NE-PER™ Nuclear and Cytoplasmic Extraction Reagents (Thermo Fisher Scientific). Protein concentrations were determined with the Micro BCA protein assay Kit (Thermo Fisher Scientific). Equal amounts of proteins (30 µg for the total lysate, 10 µg for the cytosolic fraction, and 5 µg for the nuclear fraction) were separated using 4–20% SDS-polyacrylamide gel electrophoresis and transferred onto nitrocellulose membranes using a trans-Blot Turbo System (Bio-Rad Laboratories). Membranes were blocked in 5% non-fat milk in 1X Tris-Buffered Saline, 0.1% Tween 20 Detergent (TBST) for 1 h and then incubated overnight with primary antibodies diluted in 5% non-fat milk in TBST. Primary and secondary antibodies with their dilutions are listed in Table 2. The immunoreactive bands were visualized using the Clarity Western ECL substrate (Bio-Rad Laboratories). Images were acquired using the Chemidoc XRS+ (Bio-Rad Laboratories). For quantitative band densitometry, Image Lab 3.0 software (Bio-Rad Laboratories) was used. The data obtained from the nuclear fraction were normalized to H3 histone, while the data obtained from cytosolic fraction and from total lysates were normalized to β-actin. In the case of NQO1, we obtained two closely spaced immunoreactive bands. The densitometric analysis was performed on the upper one, near the molecular weight of NQO1 protein (31 kDa).

### 2.7. 2′,7′-Dichlorofluorescein Diacetate (DCFH-DA) Assay

The DCFH-DA assay is a widely used method to detect ROS in different cells and tissues, and it has been employed previously to detect ROS in retinal explants [44]. We used it to visualize ROS in retinal explants incubated for six days in the different experimental conditions. The explants were incubated for 15 min with 1 µM DCFH-DA in saline at 37.0 ± 0.5 °C and the images were acquired using an epifluorescence microscope (Nikon Europe, Amsterdam, The Netherlands).

### 2.8. Statistics

Differences between groups were tested using unpaired t-test or one-way ANOVA followed by Newman-Keuls multiple comparison post-hoc test (GraphPad Prism 8, San Diego, CA, USA). The results were expressed as mean ± SEM of the indicated n values. Differences with *p* < 0.05 were considered statistically significant.

## 3. Results

### 3.1. Antioxidant Enzyme Expression and Nrf2 Nuclear Translocation

As a first step, we performed a dose-response analysis after 24 h incubation with NIH1 to determine the concentration of NIH1 inducing significant expression of Nrf2 itself and of antioxidant genes. In retinal explants treated with 50 µM NIH1, the expression of Nrf2, HO-1, and NQO1 mRNAs was considerably increased by about 16-, 250-, and 6-fold, respectively. Significantly high HO-1 mRNA levels were also observed with 15 µM NIH1 (Figure 2A–C). The analysis of Nrf2 protein levels in retinal explants treated with 50 µM NIH1 confirmed that the surge of antioxidant gene expression was concomitant with a significant increase in both Nrf2 cytosolic content (Appendix A) and in Nrf2 nuclear translocation, (Figure 2D). Based on these results, the subsequent experiments were conducted using 50 µM NIH1.

### 3.2. Long-Term Effects of NIH1 Treatment

As shown in Figure 3, the expression of Nrf2, HO-1, and NQO1 mRNAs was increased by about 3- to 4-fold after six days of NIH1 treatment, compared to controls (Figure 3A,C,E). This was correlated with an enhanced Nrf2 nuclear translocation, as shown by the Western blot analysis of the nuclear protein fraction (about 2.5-fold; Figure 3B). The NIH1-driven increase of HO-1 and NQO1 mRNAs correlated with the increment in their relative protein content, which was increased by 3- to 5- fold in the treatment groups with respect to controls (Figure 3D,F).

To ascertain whether the administration of NIH1 for six days could interfere with retinal cell viability and/or glial cell stability, the protein levels of cleaved caspase-3 and of GFAP, used as markers of apoptosis and of gliosis, respectively, were evaluated (Figure 4A). The results showed that a six-day administration of NIH1 did not lead to any significant changes in either marker, as the protein levels of both cleaved caspase-3 (Figure 4B) and GFAP (Figure 4C) were similar in NIH1-treated and in control explants. Confirming the cleaved caspase-3 data, also the ratio cleaved caspase-3/procaspase-3 as evaluated with Western blotting (Appendix A) and the mRNA levels of caspase-3 evaluated with qPCR were similar in control and NIH1/6d explants (Appendix A).

### 3.3. Effects of NIH1 under OS Conditions

As shown in Figure 5A,C,E, the OS/2d group did not display any significant alterations in Nrf2, HO-1, or NQO1 mRNA compared to controls. Conversely, the treatment with NIH1 for two days produced a change in Nrf2, HO-1 and NQO1 expression (NIH1 +OS/2d bars in Figure 5A,C,E), which significantly increased by 3-, 79- and 11-fold, respectively. The nuclear levels of Nrf2, were comparable to those of controls in the OS/2d group, while they were increased by 2-fold in NIH1 + OS/2d explants (Figure 5B). Similarly, the HO-1 and NQO1 protein levels were unaltered in OS/2d explants compared to controls, while they were drastically enhanced by 11- and 6-fold following treatment with NIH1 (NIH1 + OS/2d bars in Figure 5D,F).

In contrast to OS/2d explants, OS/6d explants showed an overall increase by about 2-fold in Nrf2, HO-1 and NQO1 mRNA expression, while the OS-driven antioxidant gene expression was fully counteracted in NIH1 + OS/6d explants (Figure 6A,C,E). This modulation of Nrf2 and antioxidant gene expression was consistent with the observed levels of Nrf2 nuclear protein content, which was increased by about 1.5-fold in OS/6d explants, compared to controls, and restored to basal levels after six days of treatment with NIH1 (NIH1 + OS/2d bar in Figure 6B). Interestingly, the protein levels of HO-1 and NQO1 in OS/6d and in NIH1 + OS/6d explants diverged from the relative qPCR data, since they were significantly reduced in OS/6d explants, while they were dramatically increased in NIH1 + OS/6d explants (Figure 6D,F).

To determine if these observed effects of NIH1 could lead, ultimately, to a reduction of ROS accumulation in the retina, we used DCFH-DA labeling to directly visualize the presence of ROS in the retinal tissue. As shown in Figure 7, DCFH-DA fluorescence was considerably increased in OS/6d explants compared to controls, while the treatment with NIH1 completely reversed the picture and recovered a staining pattern similar to control explants.

### 3.4. Effects of NIH1 on OS-Induced Apoptosis and Reactive Gliosis

In order to evaluate the effect of NIH1 on apoptosis and reactive gliosis in stressed retinal explants, we analyzed the protein levels of cleaved caspase-3 and of GFAP. As expected, the levels of both cleaved caspase-3 and GFAP increased significantly in OS/6d explants, while these increases were completely prevented by treatment with NIH1 (Figure 8). An almost identical pattern of changes was observed for the ratio cleaved caspase-3/procaspase-3 evaluated with Western blotting (Appendix A) and for the levels of caspase-3 mRNA expression as evaluated with qPCR (Appendix A).

## 4. Discussion

OS is a pathophysiological mechanism involved in the onset of different retinal diseases. Indeed, high ROS levels and decreased antioxidant activity make retinal tissue incline to develop inflammation, neurodegeneration, and consequent structural and functional deficits [3,7,45]. To counteract OS, different therapies have been tested, among which the use of antioxidant natural compounds. Here, using an ex vivo retinal model of OS, we demonstrated that administration of NIH1, a natural hybrid derived from the combination of functional features of curcumin and diallyl sulfide, could be a useful strategy for treating or preventing retinal diseases whose common denominator is OS.

### 4.1. NIH1 Promotes Nrf2 Activation and Antioxidant Gene Expression

Nrf2 is widely known as a major activator of the antioxidant response and it has been hypothesized that changes in Nrf2 efficacy could be involved in the onset of a variety of retinal diseases [46,47]. Indeed, impairment of Nrf2 functionality perturbs retinal cellular homeostasis causing exacerbation of OS, inflammation, and cell death [48,49,50]. Therefore, in the presence of OS, boosting the antioxidant defense system through Nrf2 activation could be a way to preserve retinal health. Several studies have highlighted that some natural compounds can activate Nrf2 favoring its nuclear translocation [15,51,52] and that the chemical manipulation of these compounds may increase their efficacy, as in the case of NIH1 [30]. Our results show that NIH1 induces Nrf2 nuclear translocation in ex vivo retinal explants, a model that very closely mimics the in vivo retina, thus marking an advancement with respect to previous in vitro studies with cell lines. In those studies, NIH1 and related compounds were assumed to exert their effects by directly interacting with Keap1 and preventing its binding to Nrf2 [28]. The same mechanism is likely to mediate NIH1 effects in the ex vivo model; however, it should be noted that Nrf2 activity is tightly regulated through both pre- and post-translational mechanisms, including, respectively, miRNAs [53] and phosphorylation by different types of kinases, which may either positively or negatively regulate Nrf2 [54]. Therefore, NIH1 may also take part in miRNA- or kinase-mediated Nrf2 regulation, although further studies will be necessary to investigate this possibility.

Our data show that NIH1-induced Nrf2 nuclear translocation is correlated with an increase of Nrf2 mRNA levels. This is consistent with data showing that, once in the nucleus, Nrf2 binds the antioxidant response element (ARE) sequences and induces its own expression via a positive feedback [55,56]. Moreover, the binding of Nrf2 to ARE also causes the transcription of phase II antioxidant enzymes, including HO-1 and NQO1 [57]. Accordingly, in retinal explants treated with NIH1, we showed an increase of HO-1 and NQO1 expression both at the mRNA and at the protein level. These results are in line with those obtained in previous studies using different cell lines. In particular, in ARPE 19 cells, NIH1 administration induced Nrf2 nuclear translocation, Nrf2 mRNA expression, and increased HO-1 levels [32]. Taken together, these data support the idea that NIH1 is a hybrid with strong antioxidant power and with no adverse effects on retinal cell viability or glial activation, as demonstrated by our data showing no changes in cleaved caspase-3 or GFAP protein levels in retinal explants incubated in the presence of NIH1 for six days. 

### 4.2. NIH1 Strengthens the Antioxidant Response and Prevents Retinal Cell Death and Glial Activation

Our results show no activation of Nrf2 nuclear translocation or increased expression of antioxidant enzymes in explants incubated for 24 h under OS. This is not entirely consistent with our previous data indicating an increase of Nrf2, NQO1, and HO-1 mRNA expression in retinal explants after 12 or 24 h OS [37,43]. This discrepancy may be explained considering that, although OS is maintained for 24 h in all cases, the OS/2d retinal explants of the present study have undergone a 24 h pre-treatment with basal culture medium. Therefore, these explants had the time to adapt to the culture environment before OS onset, and this may be the reason why their response to OS is less immediate than that of explants incubated in OS right after dissection. In summary, the overall picture in OS/2d explants is that of a tissue in which a substantial antioxidant response has yet to begin. If the OS treatment is prolonged to the 6th day, Nrf2 is activated, and the downstream antioxidant gene expression is increased. However, despite increased HO-1 and NQO1 mRNAs, the protein levels are significantly decreased. This apparent contradiction is likely to derive from an intense turnover and post-transcriptional mechanisms that influence the half-life of antioxidant enzymes in response to OS. For instance, HO-1 has been found to accumulate more into lysosomes than in the cytoplasm of retinal pigment epithelial cells in exudative AMD patients, suggesting a high enzyme turnover due to degradation induced by lysosomes [58]. Moreover, astrocytes exposed to prolonged inflammation consistently increased HO-1 mRNA expression while reducing HO-1 protein levels due to immunoproteasome-mediated degradation [59]. Similarly, NQO1 was observed to undergo polyubiquitination and then proteasome degradation, compromising its antioxidant action, in neuroblastoma cells [60].

The reduction of antioxidant enzymes aggravates the retinal oxidant status. Indeed, in OS/6d explants, the decrease of antioxidant enzymes is concomitant with a massive increase of DCFH-DA fluorescence, indicating high ROS levels. Conversely, the data obtained in NIH1 + OS/2d and in NIH1 + OS/6d explants suggest that the treatment with NIH1 ensures Nrf2 nuclear translocation and a massive expression of antioxidant enzymes since OS onset, which would allow accumulation of HO-1 and NQO1 proteins throughout the incubation period. In this context, the OS-driven depletion of antioxidant enzymes would be prevented by the early stimulation of Nrf2, thus ensuring the maintenance of a high endogenous antioxidant potential that is effective in preventing ROS accumulation, as demonstrated by DCFH-DA analysis.

Another consequence of persistently high levels of HO-1 and NQO1 proteins is the lack of HO-1 and NQO1 mRNA upregulation, as observed in NIH1 + OS/6d explants. Indeed, a sort of autoregulatory downregulation mechanism is likely to be implemented by retinal cells in response to the strong and continuous Nrf2 activation by NIH1 and the massive accumulation of HO-1 and NQO1 proteins. This interpretation is supported by experimental findings indicating that increasing Nrf2 levels generate negative feedback mechanisms, including the expression of components of the ubiquitin ligase complex involved in ubiquitination and proteasome degradation, which regulate Nrf2 degradation to avoid excessive transcriptional activity [61,62].

Together, the present investigations suggest that NIH1 may interact with Nrf2 and induce complex mechanisms that, ultimately, limit ROS accumulation in the retina, even in the presence of persistent stressing conditions. Most importantly, our data show that not only is NIH1 devoid of apparent negative side effects on retinal viability, but also demonstrate that the mechanisms triggered by its application result in protection of retinal cells from apoptosis and in prevention of glial reaction, as demonstrated by the levels of cleaved caspase-3 and of GFAP in NIH1 + OS/6d explants. In particular, in agreement with previous observations in retinal explants [37,41], the results showed that caspase-3 mRNA expression follows the changes of the cleaved caspase-3 protein, suggesting that the overall response of the retina to an oxidative insult includes both an upregulation of the transcription of the caspase-3 gene and an increased formation of the cleaved, active form of the enzyme. These observations are in line with those obtained with other antioxidant natural substances that may activate Nrf2 [11,18,63]. However, the efficacy of NIH1 suggests that the chemical manipulation of natural compounds may be an efficient strategy to search for novel candidates for the treatment of retinal diseases.

## 5. Conclusions

From the data reported in this study, we infer that NIH1 is a promising nature-inspired Nrf2 activator that may be investigated in further studies in in vivo models of DR, AMD, or glaucoma to better clarify the therapeutic potential of this Nrf2 activator.

## Figures and Tables

**Figure 1 antioxidants-10-01296-f001:**
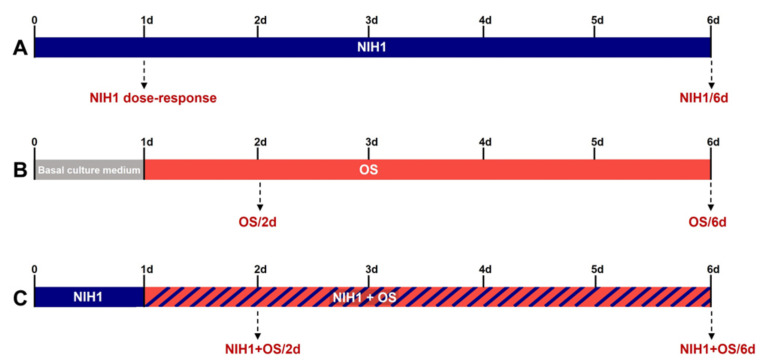
Graphical timeline describing the experimental design. The treatments are color coded: NIH1, blue; OS, orange; NIH1 + OS, blue-dashed orange, basal culture medium, gray. (**A**) NIH1 effects: Dose-response analysis performed after one day (1d) and long- term analysis performed after six days (NIH1/6d) of incubation. (**B**) OS model: Pre-treatment with basal culture medium for the first day and OS treatment for the subsequent one day (OS/2d) or five days (OS/6d) of incubation. (**C**) NIH1 effects in OS: Pre-treatment with NIH1 for the first day and NIH1 + OS treatment for the subsequent one day (NIH1 + OS/2d) or five days (NIH1 + OS/6d) of incubation.

**Figure 2 antioxidants-10-01296-f002:**
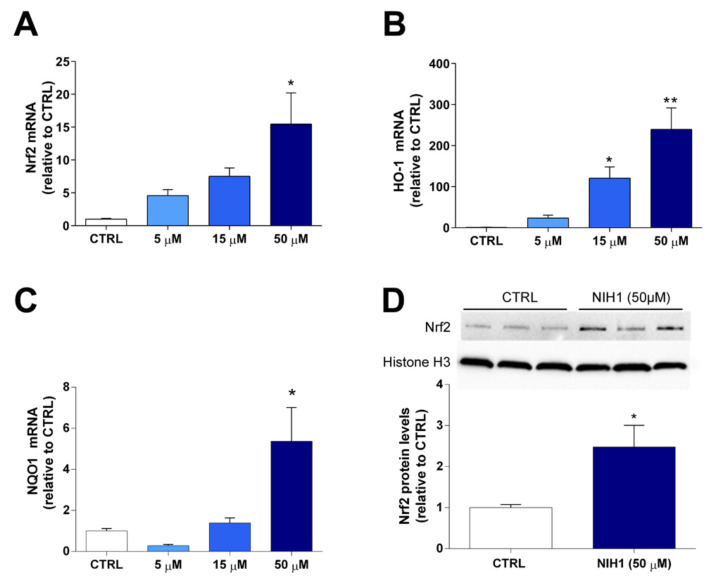
Dose-response analysis of mRNA expression and Nrf2 nuclear translocation in retinal explants treated with NIH1. The mRNA levels of Nrf2 (**A**), HO-1 (**B**), and NQO-1 (**C**) were evaluated with qPCR in control retinal explants (CTRL) or in retinal explants treated with 5, 15, or 50 µM NIH1 for 24 h. One-way ANOVA followed by the Newman-Keuls multiple comparison post-hoc test (*n* = 3; * *p* < 0.05, ** *p* < 0.01 vs. CTRL). (**D**) Western blot analysis of nuclear protein fraction showing representative immunoreactive bands and quantitative densitometric analysis of the Nrf2 protein levels in CTRL explants and in explants treated with 50 µM NIH1 for 24 h. Unpaired *t-*test, *n* = 3; * *p* < 0.05, ** *p* < 0.01 vs CTRL.

**Figure 3 antioxidants-10-01296-f003:**
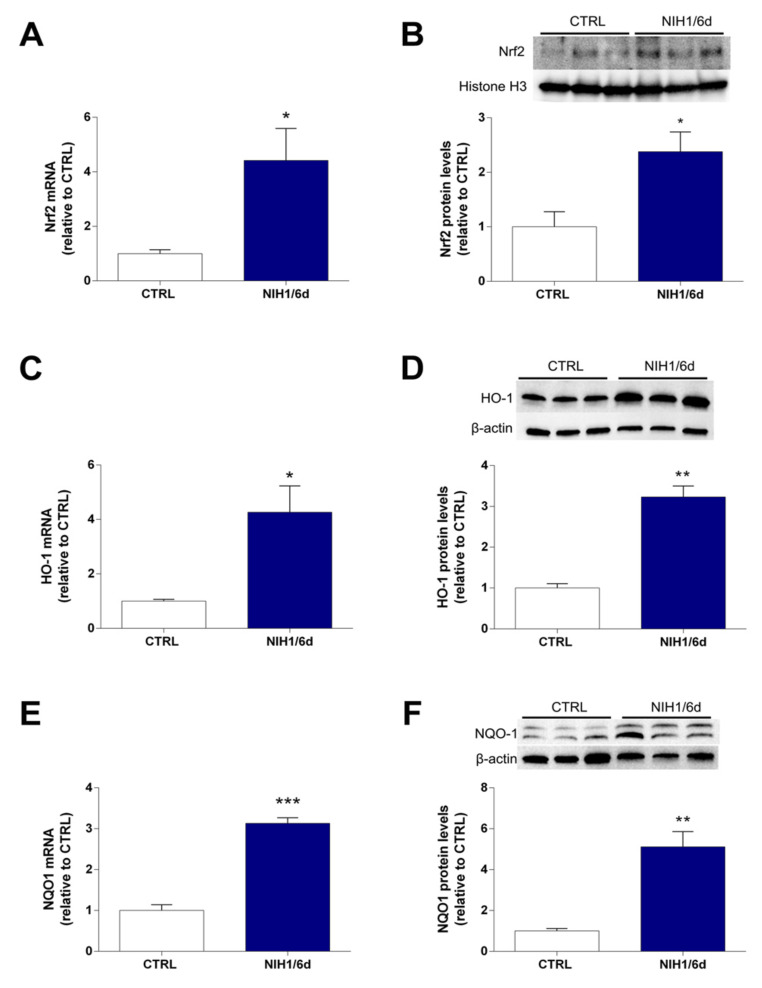
Long-term effects of NIH1 on Nrf2 activation and related antioxidant enzymes. The mRNA levels of Nrf2 (**A**), HO-1 (**C**), and NQO-1 (**E**) were evaluated with qPCR in CTRL explants and in retinal explants treated with 50 µM NIH1 for six days. (**B**) Western blot of nuclear protein fraction showing representative immunoreactive bands and quantitative densitometric analysis of the Nrf2 protein levels in CTRL and in retinal explants treated with 50 µM NIH1 for six days. (**D**,**F**) Western blot analysis of total protein fraction showing representative immunoreactive bands and quantitative densitometric analysis of the HO-1and NQO1 (respectively) protein levels in CTRL and in retinal explants treated with 50 µM NIH for six days. Unpaired t-test, *n* = 3; * *p* < 0.05, ** *p* < 0.01, *** *p* < 0.001 vs. CTRL.

**Figure 4 antioxidants-10-01296-f004:**
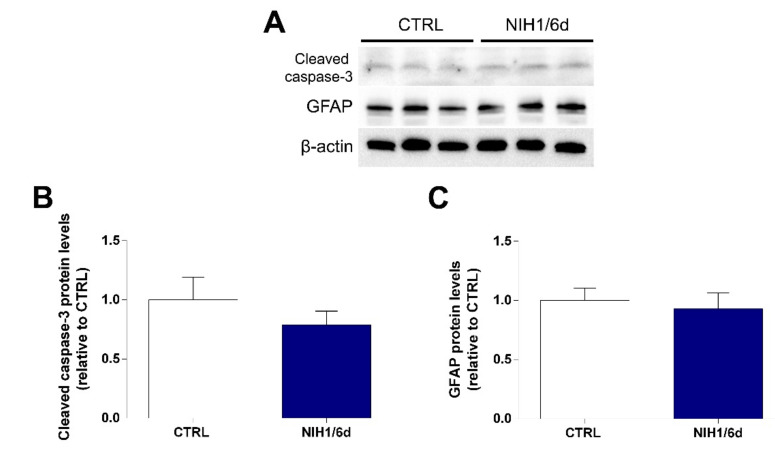
Effect of NIH1 on retinal cell viability and glial cell reactivity. Western blot analysis of total protein fraction showing representative immunoreactive bands (**A**) and quantitative densitometric analysis of the cleaved caspase-3 (**B**) and GFAP (**C**) protein levels in CTRL explants and in retinal explants treated with 50 µM NIH1 for six days. Unpaired *t*-test, *n* = 3.

**Figure 5 antioxidants-10-01296-f005:**
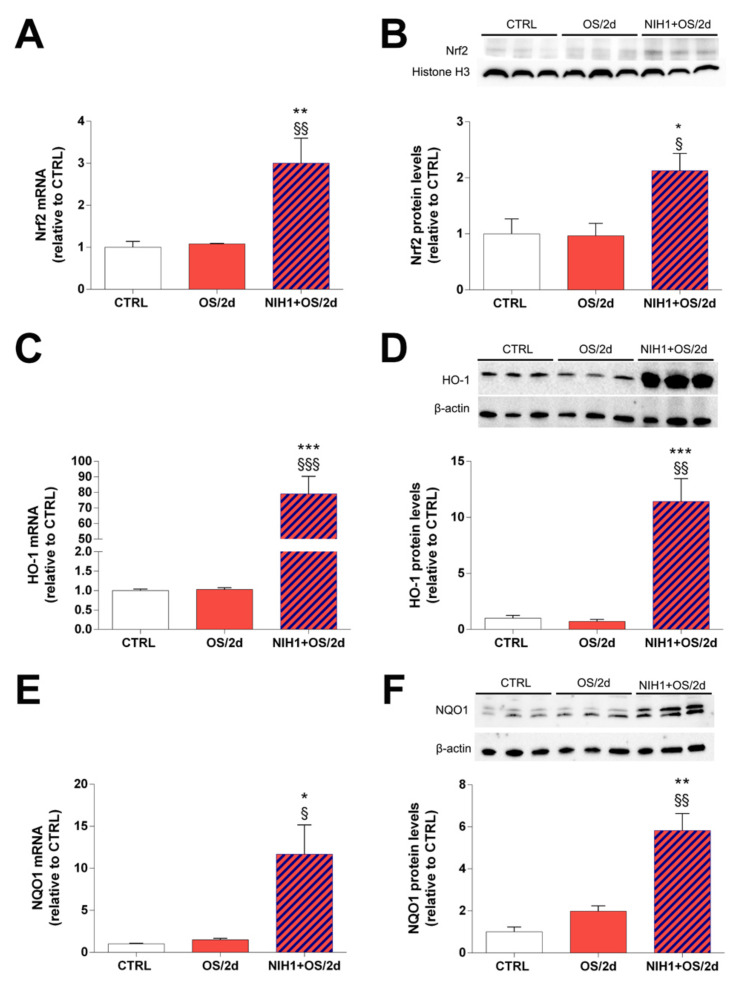
Short-term antioxidant effect of NIH1 in retinal explants treated with NIH1 for two days. The mRNA levels of (**A**) Nrf2, (**C**) HO-1, and (**E**) NQO-1 were evaluated with qPCR in CTRL explants, in retinal explants treated with OS for two days (OS/2d), and in retinal explants treated with OS together with NIH1 for two days (NIH1 + OS/2d). (**B**) Western blot analysis of nuclear protein fraction showing representative immunoreactive bands and quantitative densitometric analysis of the Nrf2 protein levels in CTRL, OS/2d and NIH1 + OS/2d explants. (**D**,**F**) Western blot analysis of total protein fraction showing representative immunoreactive bands and quantitative densitometric analysis of the HO-1 and NQO1 protein levels in CTRL, OS/2d and NIH1 + OS/2d explants. One-way ANOVA, *n* = 3; * *p* < 0.05, ** *p* < 0.01, *** *p* < 0.001 vs. CTRL; ^§^
*p* < 0.05, ^§§^
*p* < 0.01, ^§§§^
*p* < 0.001 vs. OS/2d.

**Figure 6 antioxidants-10-01296-f006:**
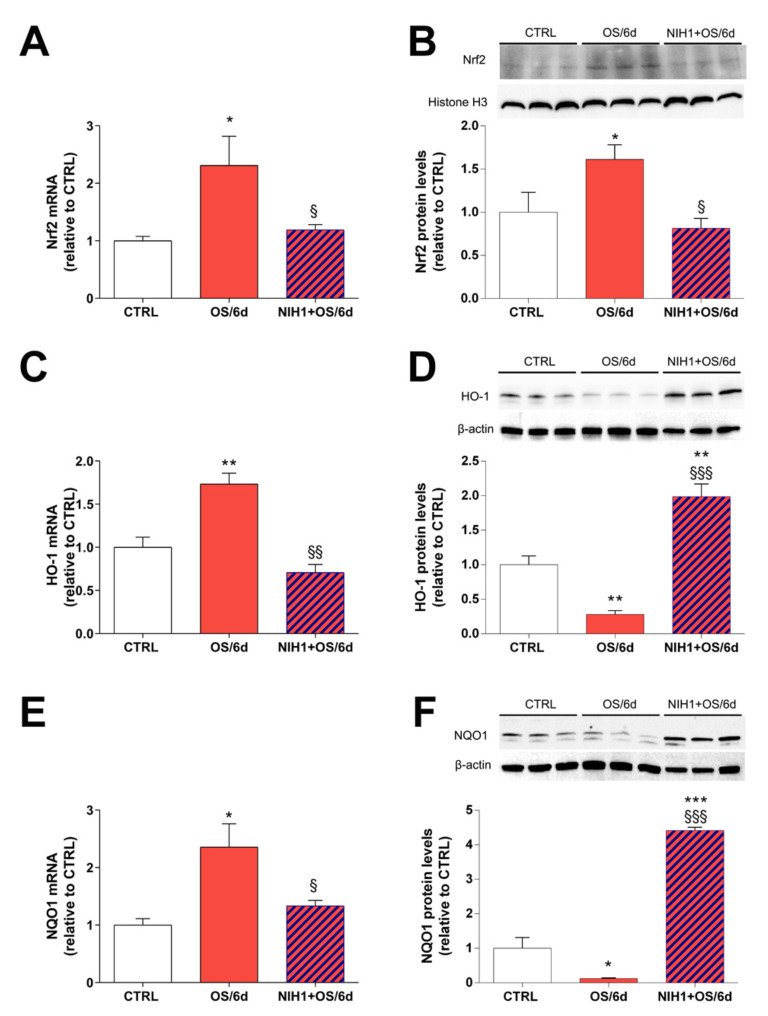
Long-term antioxidant effect of NIH1 in retinal explants treated with NIH1 for six days. The mRNA levels of (**A**) Nrf2, (**C**) HO-1, (**E**) NQO-1 were evaluated with qPCR in CTRL, in OS/6d, and in NIH1 + OS/6d explants. (**B**) Western blot analysis of nuclear protein fraction showing representative immunoreactive bands and quantitative densitometric analysis of the Nrf2 protein levels in CTRL, OS/6d and NIH1 + OS/6d explants. (**D**,**F**) Western blot analysis of total protein fraction showing representative immunoreactive bands and quantitative densitometric analysis of the HO-1and NQO1 protein levels in CTRL, OS/6d and NIH1 + OS/6d explants. One-way ANOVA, *n* = 3; * *p* < 0.05, ** *p* < 0.01, *** *p* < 0.001 vs. CTRL; ^§^
*p* < 0.05, ^§§^
*p* < 0.01, ^§§§^
*p* < 0.001 vs. OS/6d.

**Figure 7 antioxidants-10-01296-f007:**
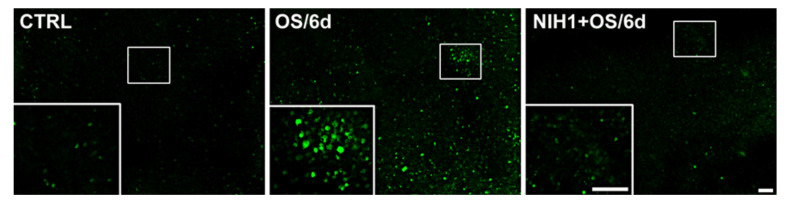
Effect of NIH1 on ROS levels. Visualization of ROS in CTRL, OS/6d, and NIH1 + OS/6d explants using the DCFH-DA assay. Insets are higher-power images of the boxed areas. Scale bar, 100 µm (50 µm for the insets).

**Figure 8 antioxidants-10-01296-f008:**
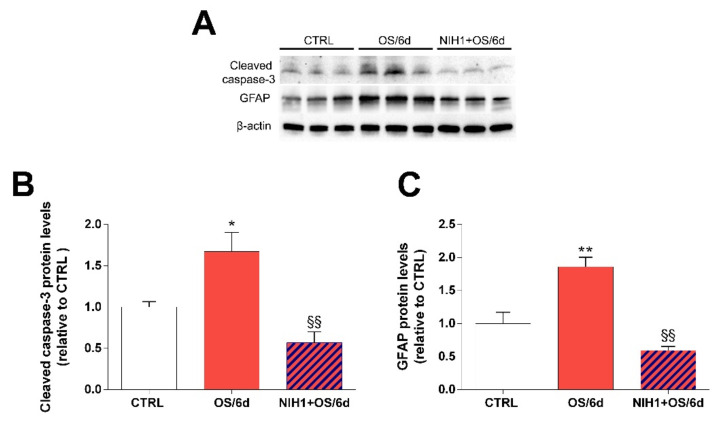
Effect of NIH1 on retinal cells apoptosis and glial reactivity. Western blot analysis of total protein fraction showing representative immunoreactive bands (**A**) and quantitative densitometric analysis of the cleaved caspase-3 (**B**) and GFAP (**C**) protein levels in CTRL, OS/6d and NIH1 + OS/6d explants. One-way ANOVA, *n* = 3; * *p* < 0.05, ** *p* < 0.01 vs. CTRL; ^§§^
*p* < 0.01 vs. OS/6d.

**Table 1 antioxidants-10-01296-t001:** Primer sequences used for qPCR.

Gene	Forward 5′-3′	Reverse 5′-3′	Accession No.
*Nrf2*	TCTTGGAGTAAGTCGAGAAGTGT	GTTGAAACTGAGCAAAAAAGGC	NM_010902.4
*HO-1*	AAGCCGAGAATGCTGAGTTCA	GCCGTGTAGATATGGTACAAGGA	NM_010442.2
*NQO1*	AGGATGGGAGGTACTCGAATC	AGGCGTCCTTCCTTATATGCTA	NM_008706.5
*Caspase-3*	GCACTGGAATGTCATCTCGCTCTG	GCCCATGAATGTCTCTCTGAGGTTG	NM_009810.3 NM_001284409.1
*Rpl13A*	CACTCTGGAGGAGAAACGGAAGG	GCAGGCATGAGGCAAACAGTC	NM_009438.5

**Table 2 antioxidants-10-01296-t002:** Antibodies used for the Western blotting.

Antigen	Dilution	Type of Ab	Source	Catalog No.
Nrf2	1:400	Rabbit monoclonal	Abcam	ab62352
GFAP	1:500	Rabbit monoclonal	Abcam	ab207165
HO-1	1:500	Rabbit polyclonal	Abcam	ab13243
NQO1	1:500	Rabbit polyclonal	Abcam	ab34173
Cleaved caspase-3	1:500	Rabbit monoclonal	Cell Signaling Technology	9664
Cleaved Caspase-3 *	1:500	Rabbit polyclonal	Cell Signaling Technology	9661
H3 histone	1:2500	Rabbit monoclonal	Abcam	ab1791
β-actin	1:2500	Mouse monoclonal	Sigma-Aldrich	A2228
Rabbit IgG HRP ** conjugate	1:5000	Goat polyclonal	Bio-Rad	1706515
MouseIgG HRP * conjugate	1:5000	Rabbit polyclonal	Sigma-Aldrich	A9044

* This antibody labels both procaspase-3 and cleaved caspase-3 (antibody data sheet). ** HRP, horseradish peroxidase.

## Data Availability

The data presented in this study are available on request from the corresponding authors.

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
