# Peer review of "A Nature-Inspired Nrf2 Activator Protects Retinal Explants from Oxidative Stress and Neurodegeneration"

_antioxidants, 2021, doi:10.3390/antiox10081296_

Round 1

Reviewer 1 Report

The manuscript by Rossino et al describes the anti-oxidant properties of a nature-inspired hybrid molecule (NIH1). In particular, the author shows that NIH1 activates the Nrf2 pathway in organotypic retinal explants.

Comments:

The anti-oxidant properties of NIH1 were already demonstrated in different cell lines and the originality of the paper is based on the use of the ex-vivo model that mimics the retina. For this reason, this model needs to be better characterized: the method “Ex-vivo model” is very poor and no references are indicated. Data on explant morphology and viability are missing. To assess the preservation of the retinal microarchitecture over time also in the absence of oxidative stress, the authors should stained explant sections with the nuclear marker DAPI and with anti-neuronal markers antibodies.

As NIH1 is synthesized by the authors, more details about its synthesis and characterization are needed at least in Supplemental Materials.

Please, add figure numbers when results are mentioned within the Discussion.

Author Response

The anti-oxidant properties of NIH1 were already demonstrated in different cell lines and the originality of the paper is based on the use of the ex-vivo model that mimics the retina. For this reason, this model needs to be better characterized: the method “Ex-vivo model” is very poor and no references are indicated. Data on explant morphology and viability are missing. To assess the preservation of the retinal microarchitecture over time also in the absence of oxidative stress, the authors should stained explant sections with the nuclear marker DAPI and with anti-neuronal markers antibodies.

R - We thank the Reviewer for this comment. Although the effect of NIH1 has been demonstrated previously in in vitro models, its efficacy on the whole organ in an ex vivo model with a higher grade of functional and structural complexity is not obvious. In effect, cases of loss of treatment efficacy when passing from cell culture to whole organ are not rare. Therefore, we believe that the strength of the present study is not limited to expanding the evidence of NIH1 efficacy on a model more closely related to a “real” retina but, mainly, it paves the way for the translation of NIH1 treatment in vivo. We have used the ex vivo model of retinal explants in the presence of stressors such as oxidative stress for years, by exploiting its high propaedeutic potential for subsequent studies in vivo for basic research and pharmacological investigations.

As the Reviewer pointed out, we acknowledge that, as presented, the characterization of the ex vivo model in the methods appears poor and not supported by previous evidence. We did not provide an exhaustive characterization of the ex vivo model because it has been reported in our previous publications. In particular, the DAPI staining requested by the reviewer has been provided in the study reported in reference 41. Regarding the anti-neuronal markers, we have immunolabeled different types of retinal neurons (ganglion cells, bipolar cells, amacrine cells) in another paper (reference 42), while further studies showing structural features and molecular mechanisms triggered by oxidative stress in our ex vivo model are in references 37 and 43.  In agreement with the observation of the Reviewer, we added these supporting references in the methods (lines 97 and 109-111).

As NIH1 is synthesized by the authors, more details about its synthesis and characterization are needed at least in Supplemental Materials.

R – The details of NIH synthesis and characterization have been published in previous papers, as indicated (references 27 and 28). However, we agree with the Reviewer that a short summary of the procedure may be useful, and we have included it in the new paragraph 2.3.

Please, add figure numbers when results are mentioned within the Discussion.

R - Sorry, but in our papers we have never been used to recall specific figures of the results in the Discussion, and we would like to maintain our style.

Reviewer 2 Report

Major revision

This paper Maria Grazia Rossino et al demonstrated that protective effects of NIH1, [S-allyl (E)-3-(3,4-dihydroxyphenyl)prop-2-enethioate], on Oxidative stress-induced retinal damage in mouse retinal explants. They also showed that NIH1 trigger an antioxidant response through activation of the Nrf2/Keap1-dependent pathway. The general purpose of this study is clear. The study appears to be of interest, whereas the quality of presentation has some problems. Especially, I feel the quality of figure is poor. In my opinion it requires major revision before ready for publication.

  1. Introduction or Discussion

Why did you focus on the GFAP positive cells? I recommend you to add more information of the GFAP positive cells. Are that the astrocyte and/or Müller cell in mouse retia? The authors should explain or discuss this point in the revised manuscript. I recommend you to add more information of GFAP positive cells in the Introduction section.

  1. Data

Why did not you show the levels of Nrf2 in total protein fraction? Because the mRNA levels of Nrf2 is upregulated in the NIH1-treated cells, I recommend you to add more data of the expression levels of Nrf2 in cytosolic fraction.

  1. Fig3 and5

In Western blot, two band can be seen Fig3A and Fig5F. The specific band of NQO-1 is upper one or lower one?

Minor points

  1. Uniform an abbreviation of character.

Ex) h and 1 hour, β-actin and β actin, NOQ-1 and NQO-1

I hope that my comment is very useful for the improvement of the article.

Author Response

  1. Introduction or Discussion

Why did you focus on the GFAP positive cells? I recommend you to add more information of the GFAP positive cells. Are that the astrocyte and/or Müller cell in mouse retia? The authors should explain or discuss this point in the revised manuscript. I recommend you to add more information of GFAP positive cells in the Introduction section.

R - In the introduction, a few lines (81 - 87) have been added to better explain that GFAP was used as a recognized marker of inflammation and macroglial activation. A considerable number of papers in the literature has reported increased GFAP (evaluated with different techniques) in astrocytes and / or Muller cells in diseased retinas. In most circumstances, GFAP upregulation has been observed both in astrocytes and in Muller cells. In particular, astrocytes are mostly involved in GFAP upregulation when oxidative stress involves the inner retinal layers, as in glaucoma. In contrast, GFAP upregulation is most evident in Muller cells when oxidative stress also impacts the outer retina, as in diabetic retinopathy (see, for instance reference 35). In our model, the treatment with H2O2 is likely to have a similar impact in both inner and outer retina, therefore it is plausible to imagine that both astrocytes and Muller cells are similarly involved. In any case, in the present study we used a technique (Western blotting) that does not allow to distinguish between different cell types. In addition, the specific localization of GFAP upregulation would not seem to add relevant information to the paper. Therefore, we preferred to keep the analysis of the GFAP levels as an index showing protective effects of the treatment with NIH1 without getting into deep discussions about the different macroglial cell types that might be involved.

  1. Data

Why did not you show the levels of Nrf2 in total protein fraction? Because the mRNA levels of Nrf2 is upregulated in the NIH1-treated cells, I recommend you to add more data of the expression levels of Nrf2 in cytosolic fraction.

R - We followed this constructive suggestion of the Reviewer and provided the data of cytosolic Nrf2 protein levels in NIH1-treated explants at 24 h incubation (Fig. S1). Therefore, these data confirm that Nrf2 mRNA increase is correlated to an increase of Nrf2 cytosolic protein.

  1. Fig3 and 5

In Western blot, two band can be seen Fig3A and Fig5F. The specific band of NQO-1 is upper one or lower one?

The Reviewer’s observation is correct. The Abcam Anti-NQO1 antibody (ab34173) datasheet reports that both the predicted and the observed band size corresponds to 31 kDa. However, as observed in other mouse tissues (PMID: 32997272; PMID: 32630085), incubation with the same NQO1 primary antibody could generate two close labeled bands. We compared the molecular weight of the bands with prestained recombinant proteins (10 - 250 kDa), establishing that the specific band corresponding to NQO-1 is the upper one.  For this reason, the densitometric analysis was performed considering only the upper bands. These bands were indicated with an arrow in the full blots pdf submitted together with the manuscript. We specified this point in paragraph 2.6 (lines 188-190)

Minor points

  1. Uniform an abbreviation of character.

Ex) h and 1 hour, β-actin and β actin, NOQ-1 and NQO-1

R - We apologize for these discrepancies. The abbreviations have been uniformed.

Reviewer 3 Report

This is a well-written paper on an interesting topic. The design and methods are straight-forward. Specific comments are below.

  1. In the Introduction the authors fail to mention another way that Nrf2 is regulated - by phosphorylation. There is a recent review that covers this (Liu et al., FRBM, 2021, doi: 101016/j.freeradbiomed.2021.03.034) and probably other papers too. Please add this mechanism into the Introduction.
  2. Results: Authors should add a western blot for phosphorylated Nrf2 and/or assess oxidation of Keap 1 to help explain their results.
  3. Results, Figures 4 and 8: Authors need to add Western blots and quantification of total caspase 3 and then quantify the ratio of total:cleaved caspase 3 in all conditions.

Author Response

  1. In the Introduction the authors fail to mention another way that Nrf2 is regulated - by phosphorylation. There is a recent review that covers this (Liu et al., FRBM, 2021, doi: 101016/j.freeradbiomed.2021.03.034) and probably other papers too. Please add this mechanism into the Introduction.

R - We thank the Reviewer for this suggestion. Indeed, the regulation of Nrf2 is quite a complex matter. Nature inspired hybrids, among which NIH1, were designed to interact with Keap1 (ref 28). For the sake of simplicity, we preferred not to analyze the details of the possible mechanisms for Nrf2 regulation in the introduction. However, this is an important point and we added some lines in the discussion to acknowledge the possibility that NIH1 may take part both in miRNA and in kinase-mediated Nrf2 regulation (lines 340-350).

  1. Results: Authors should add a western blot for phosphorylated Nrf2 and/or assess oxidation of Keap 1 to help explain their results.

R - The new experiments proposed by the Reviewer would certainly provide very interesting information about the mechanism(s) by which NIH1 interacts with and regulates Nrf2. However, the pattern of possible Nrf2 phosphorylations and the different enzymes involved, together with the different effects that may be obtained (positive or negative Nrf2 regulation) make up a very complex picture. These investigations may be performed in one or more dedicated studies, which would probably require, at least in the first steps, the use of simpler experimental setups, therefore in vitro models in addition or instead of ex vivo retinal explants. And, in the end, the elucidation of these mechanisms, although important in perspective, was not the scope of this study, which is centered on the demonstration that a novel, nutraceutical-based compound may exert powerful antioxidant effects and save retinal neurons from death in a condition (oxidative stress) that characterizes virtually all the major retinal diseases.

  1. Results, Figures 4 and 8: Authors need to add Western blots and quantification of total caspase 3 and then quantify the ratio of total:cleaved caspase 3 in all conditions.

R - This is an interesting point. For the assessment of the apoptotic activity in the retina, in the literature the evaluation of the cleaved/pro-caspase 3 ratio is as common as the exclusive analysis of the levels of the cleaved caspase 3. In our experience, the levels of cleaved caspase 3 always correlated with other parameters of tissue viability such as inflammation, autophagy, retinal morphology, and electrophysiological assessment of retinal activity in ex vivo and in in vivo models (references 37 and 42), thus demonstrating a consolidated reliability of the levels of cleaved caspase 3 as a measure of the apoptotic activity.

Noteworthy, in our previous studies, we have also demonstrated that the increase in active caspase 3 in retinal explants often correlates with increased caspase 3 mRNA levels, suggesting a possible upregulation of the pro-enzyme (references 37 and 41). To confirm this possibility, in the present study we performed a qPCR for quantifying caspase 3 mRNA expression in all the experimental conditions. As expected, the results confirm that caspase 3 mRNA expression follows the changes of the cleaved caspase 3 protein, suggesting the possibility that the overall response of the retina to an oxidative insult includes both an upregulation of the transcription of the caspase 3 gene and an increased formation of the cleaved, active form of the enzyme. For the sake of completeness, we added these data in supplementary Figures 2 and 3. Some addition in the text have been made both in the results (lines 251-253 and314-315) and in the discussion (lines 412-417). Of course, we are aware that the qPCR analysis should be complemented with a Western Blot analysis of the pro-caspase 3 protein levels, but, unfortunately, to fulfill this aim we should run another series of experiments, which would be very difficult in the ten days allowed for revision of the manuscript and would extend considerably the number of animals involved in the study. We genuinely believe that this analysis, as presented, conveys reliable information about the changes of apoptotic activity that intervene in the different experimental conditions.

Round 2

Reviewer 1 Report

The authors addressed all my questions. 

Author Response

Thank you for your suggestions.

Reviewer 3 Report

The authors have not addressed any of the concerns raised during previous review.

  1. The authors state that they added information on kinase-mediated Nrf2 regulation to lines 340-350, however the reviewer does not see the information there.

  1. The authors have not added any requested mechanistic data to link NIH1 to Nrf2 nuclear translocation. A simple western blot for phosphorylated NRF2 would not be difficult and would provide at least some insight regarding mechanism. For this level of journal some mechanism should be included. The authors have suggested that additional experiments investigating how Nrf2 is phosphorylated (if phosphorylation is detected) and that is not true. That certainly could be left for a future paper.

  1. Regardless of what the authors were able to get published previously, quantifying cleaved protein alone is non-informative without normalizing to total levels of the protein. Further, mRNA and protein levels do not always match. The ratio of cleaved:total caspase is needed. The same homogenate should be used – no additional animals needed.

Author Response

  1. The authors state that they added information on kinase-mediated Nrf2 regulation to lines 340-350, however the reviewer does not see the information there.

R – With respect to the previous version of the manuscript, these lines were added and they are in red characters. The Reviewer is right if he/she says that not all these lines are concerned with kinase-mediated Nrf2 regulations (and we apologize for the inaccurate indication), but if he/she pays a particular attention to lines 347 – 352 (present version of the manuscript, highlighted in yellow), he/she may find the following considerations: “… however, it should be noted that Nrf2 activity is tightly regulated through both pre- and post-translational mechanisms, including, respectively, miRNAs [54] and phosphorylation by different types of kinases, which may either positively or negatively regulate Nrf2 [55]. Therefore, NIH1 may also take part in miRNA- or kinase-mediated Nrf2 regulation, although further studies will be necessary to investigate this possibility”. This represents an equilibrate (in our opinion) acknowledgement that Nrf2 may be regulated by different mechanisms, which include not only kinase-mediated phosphorylations, but also miRNAs.

  1. The authors have not added any requested mechanistic data to link NIH1 to Nrf2 nuclear translocation. A simple western blot for phosphorylated NRF2 would not be difficult and would provide at least some insight regarding mechanism. For this level of journal some mechanism should be included. The authors have suggested that additional experiments investigating how Nrf2 is phosphorylated (if phosphorylation is detected) and that is not true. That certainly could be left for a future paper.

R – We do not understand what the Reviewer says that “is not true”, but we are glad that the Reviewer recognizes that a detailed analysis of Nrf2 regulation by NIH1 deserves a dedicated investigation. In any case, the point is that we do not agree with the Reviewer on this repeated request for “a simple Western blot”. It would be a simple thing to do, but we are convinced that this would not help reaching the aim of the paper (which is to demonstrate NIH1 efficacy in possible treatments of retinal diseases), nor it could add relevant information about mechanistic aspects concerning NIH1 interactions with Keap1, Nrf2 and/or phosphorylating enzymes. The regulatory mechanisms involving Nrf2 are too complex to be investigated with a simple Western blot, and we will try to convince the Reviewer with the following considerations:

a) NIH1 has been synthesized to increase the efficacy of curcumin on Nrf2 modulation.

Curcumin is nearly insoluble in water, has a short half-life and a low bioavailability. After oral administration it is rapidly metabolized in the intestine and in the liver and then excreted in feces. It is chemically instable at physiological pH and photodegradable, making its handling complicated and restricting its applications (Anand et al., 2007, DOI: 10.1021/mp700113r). To improve curcumin therapeutic efficacy, new strategies have been tested: new formulations, changes in administration modalities, nanotechnology-based delivery systems, and the hybridization approach. The idea behind the synthesis of NIH1 was to optimize the exposure of chemical features that are known to be responsible for the interaction with Nrf2, such as the hydroxycinnamoyl group, a recurring chemical function of polyphenols. To this aim, starting from the first hybrid generated by merging structural moieties from curcumin and diallyl sulfide, a detailed structure-activity relationship study was performed by systematically modifying the aryl substitution pattern, the thioester function, and the aliphatic skeleton with the aim of strategically tuning the pharmacological profile (Serafini et al., 2019, DOI: 10.3389/fphar.2019.01597). As for curcumin, electrophilic features were shown to be responsible for Nrf2 activation, suggesting that NIH1 and curcumin are likely to share the same mechanisms of action on Nrf2. And, as detailed below, there is information available on these mechanisms.

b) Curcumin has been shown to bind Keap1.

There are data in the literature supporting a direct interaction of NIH1 with Keap1. Indeed, NIH1 was designed to interact primarily with Keap1, since it has been demonstrated that curcumin binds to Keap1 and determines Nrf2 activation. In particular, Cysteine residues Cys151, Cys273, and Cys288 of Keap1 are targets of oxidants, which induce conformational changes and disruption of the interaction between Nrf2 and Keap1 leading to inhibition of Nrf2 polyubiquitination (Canning et al., 2015, DOI: 10.1016/j.freeradbiomed.2015.05.034). Mass spectrometry data have demonstrated that curcumin binds to Cys151 of Keap1, while curcumin treatment of cells with mutations in the mentioned residues could not activate Nrf2 translocation (Shin et al., 2020, DOI: 10.1016/j.bcp.2020.113820; Rahban et al., 2020, DOI:10.3390/antiox9121228). In addition to curcumin, also diallyl sulfide, the other moiety that inspired the synthesis of NIH1, has been described to bind Keap1, specifically on Cys288 and the data indicate that this interaction participates to Nrf2 activation (Kim et al., 2014, DOI: 10.1371/journal.pone.0085984). These data clearly demonstrate that the most plausible mechanism by which NIH1 induces Nrf2 activation is through direct interaction with Keap1.

c) There are different phosphorylation sites in the Nrf2 molecule (see for review Liu et al., 2021, DOI: 10.1016/j.freeradbiomed.2021.03.034). For instance, PKC may directly phosphorylate Nrf2 Ser40 causing dissociation from Keap1, or GSK-3 may phosphorylate Nrf2 on residues Ser335 and Ser338 causing Nrf2 inhibition. There are multiple kinases that may phosphorylate Nrf2 at different residues, and it would be difficult to choose one of those to test possible effects of NIH1: a negative result in a simple Western blot would not exclude the possibility that NIH1 may induce Nrf2 phosphorylation at the level of other residues.

d) Curcumin has been shown to stimulate the Nrf2/ARE pathway through PKCδ activation.

Previous data in the literature have shown that curcumin may induce phosphorylation-dependent Nrf2 activation. In particular, in human monocytes curcumin has been demonstrated to activate Nrf2 and antioxidant enzyme expression in a PKCδ – dependent manner (Rushworth et al., 2006, DOI: 10.1016/j.bbrc.2006.01.065). Interestingly, recent observations demonstrate that curcumin-induced PKCδ activation does not result in Nrf2 phosphorylation but in phosphorylation of p62 at Ser351. As a consequence, phosphorylated p62 would disrupt the interaction between Keap1 and Nrf2, thus allowing Nrf2 activation. The phosphorylation of p62 seems to be crucial for curcumin-induced Nrf2 activation, since curcumin effects were abolished in p62 KO cells (Park et al., 2021, DOI: 10.1038/s41598-021-87225-8). On the basis of these findings, it would make more sense looking for a NIH1-induced phosphorylation of P62, more than of Nrf2.

e) Curcumin and GSK3β.

The possibility exists that curcumin may positively regulate Nrf2 promoting GSK3β inhibition by phosphorylation. Indeed, GSK3β is involved in the phosphorylation of Neh6 domain of Nrf2 and causes Keap1-independent Nrf2 proteasomal degradation (see review by Liu et al., 2021, DOI: 10.1016/j.freeradbiomed.2021.03.034). It has been reported that curcumin may be effective in promoting GSK3β phosphorylation and inhibition (Di Martino et al., 2020, DOI: 10.1021/acschemneuro.0c00363; Li et al., 2020, DOI: 10.3389/fbioe.2020.00625). In effect, phosphorylated GSK3β has been found to increase in curcumin-treated neuronal cells, however further observations seemed to exclude a relevant role of the signaling pathway involving GSK3β in Nrf2 activation by curcumin (Park et al., 2021, DOI: 10.1038/s41598-021-87225-8).

f) The data reported above indicate that the activating effects of curcumin on Nrf2 are due to direct binding of curcumin to Cys151 of Keap1 or to indirect p62 phosphorylation consequent to curcumin-induced PKCδ activation, which results in disruption of the interaction between Keap1 and Nrf2. Since different experiments reported that curcumin induced Nrf2 activation is abolished either if Keap1 Cys151 is mutated or if p62 is silenced, it follows that curcumin positively modulates Nrf2 through a dual action that directly or indirectly affects Keap1 and its binding to Nrf2. In addition, only little effects, if any, of curcumin have been observed on GSK3β. It is our opinion that very unlikely this picture, deriving from studies with cell lines, could obtain significant integrations or new elements from a simple Western blot in retinal explants.

g) It wouldn’t be easy to justify why we tested possible Nrf2 modulation through phosphorylation and not the modulation occurring through interactions with miRNAs.

Epigenetic regulation of Nrf2, including modifications in miRNA expression, operated by different types of phytochemicals has been reported in several studies (Bhattacharjee and Dashwood, 2020, DOI: 10.3390/antiox9090865), and curcumin has been described to act as an epigenetic regulator capable of interacting with miRNAs (Hassan et al., 2019, DOI: 10.3389/fgene.2019.00514). There is evidence of miRNA-mediated curcumin regulation of Nrf2. In particular, a curcumin analog was reported to upregulate Nrf2 by increasing miR-200a and exert protective effects in diabetic nephropathy by inhibiting miR-21 (Wu et al., 2016, DOI: 10.1007/s00125-016-3958-8), while observations in cultured tilapia hepatocytes suggest that miR-153b, miR-200a, and miR-29 are involved in the regulation of the Nrf2-Keap1 signaling pathway by curcumin (Li et al., 2021, DOI: 10.1016/j.aquatox.2021.105896).

h) Other possible types of epigenetic regulation.

Curcumin has been reported to be involved in different types of epigenetic modulation. In particular, the epigenetic regulatory roles of curcumin include, in addition to miRNA regulation, the inhibition of DNA methyltransferases, regulation of histone modifications via the regulation of histone acetyltransferases, and histone deacetylases, action as a DNA binding agent, and interaction with transcription factors (Hassan et al., 2019, DOI: 10.3389/fgene.2019.00514). Regarding possible roles of curcumin in epigenetic regulation of Nrf2, curcumin has been reported to reverse the methylation status of the first five hypermethylated CpG islands in the Nfe2l2 promoter, thereby restoring epigenetically-silenced Nrf2, in tumor cells. In addition, it seems to have negligible effects on DNA methyltransferases at the RNA or protein level, but it is effective in inhibiting their enzymatic activity (Khor et al., 2011, DOI: 10.1016/j.bcp.2011.07.065).

i) It appears that Nrf2 regulation by phosphorylation is only a part of a quite complex array of mechanisms, and curcumin may be involved in all of them. We honestly think that a simple Western blot of one of the possible phosphorylated forms of Nrf2 would only suggest to the readers of our article that our knowledge of this field is less than adequate.

  1. Regardless of what the authors were able to get published previously, quantifying cleaved protein alone is non-informative without normalizing to total levels of the protein. Further, mRNA and protein levels do not always match. The ratio of cleaved:total caspase is needed. The same homogenate should be used – no additional animals needed.

R – It is not totally clear what the Reviewer is suggesting: total caspase would mean the sum of procaspase plus cleaved protein, but in the literature we have mostly found examples of calculations of the ratio cleaved caspase-3 / procaspase-3. In any case, our papers have been published in respected journals with qualified editorial boards and reviewers. Specifically, one of these reviewers, in revising one of our papers, suggested to evaluate the level of apoptosis with Western blots of cleaved caspase-3. In effect, although we agree with the Reviewer that the evaluation of the indicated ratio might be used, we do not agree (and our opinion seems to be shared by many others) that the quantification of the cleaved protein alone is non-informative.

In particular, a rapid PubMed search using “caspase-3” and “retina” as keywords retrieved 107 results in 2020 - 2021, and we analyzed the first 50 papers using caspase-3 as an apoptotic marker. The most used methods included measurements of caspase-3 activity (5 papers), immunocytochemistry for cleaved caspase-3 (7 papers), or immunocytochemistry for procaspase-3 (2 papers). In the remaining 36 papers, caspase was evaluated with Western blotting. In 17 of these papers, data were presented of only cleaved caspase-3; in 11 only of procaspase-3; in 4 of cleaved caspase-3 together with those of procaspase-3 (but without calculating any ratio), and in 4 of the ratio cleaved caspase-3/procaspase-3. Of note, in only one of these last 4 papers, the term “total caspase-3” was used in place of “procaspase-3”.

We also performed another search in the literature: this time we checked the papers cited together with the data sheet of antibody Cell Signaling Technology #9661, which recognizes both procaspase-3 and cleaved caspase-3, and that we used to provide the data of figures S2 and S4. We wanted to see in how many papers the researchers, having the availability of an antibody that would easily allow the calculation of the ratio, effectively calculated this ratio. Among a total of 65 papers, 43 evaluated cleaved caspase-3 alone, 14 evaluated both forms and calculated the ratio, 8 evaluated both forms without calculating any ratio.

Considering these evaluations of the literature, it seems that the majority of researchers involved in these issues (and of the reviewers who revised these papers) think that reliable information on caspase-3 activation can be obtained even without calculating the mentioned ratio.

Nevertheless, we acknowledge the fact that the addition of these data (although not necessary) may be of some interest. Therefore, the analysis of the ratio cleaved caspase-3/procaspase-3 has been summarized in two new supplementary figures (S2 and S4) and in the text highlighted in yellow. The new data relative to figure S2 have been obtained using leftover homogenate from previous experiments. The new data relative to figure S4 have been obtained using a TBS-stored membrane from previous experiments following protein stripping and re-incubation with an antibody directed to caspase-3 (Cell Signaling #9661), which, as mentioned above, recognizes both procaspase-3 and cleaved caspase-3.

Round 3

Reviewer 3 Report

The authors have addressed the most important issues with the manuscript.